# High Temperature Mechanical Properties of a Vented Ti-6Al-4V Honeycomb Sandwich Panel

**DOI:** 10.3390/ma13133008

**Published:** 2020-07-06

**Authors:** Lei Shang, Ye Wu, Yuchao Fang, Yao Li

**Affiliations:** 1Center for Composite Material, Harbin Institute of Technology, Harbin 150001, China; shanglei420@163.com; 2Jiangxi Province Key Laboratory of Hydraulic and Civil Engineering Infrastructure Security, School of Civil and Architecture Engineering, Nanchang Institute of Technology, Nanchang 330029, China; 0224141@163.com; 3State Key Laboratory of Advanced Welding and Joining, Harbin Institute of Technology, Harbin 150001, China; yuchao.fang@outlook.com

**Keywords:** vented Ti-6AL-4V honeycomb sandwich panel, brazing, high temperature, mechanical behavior

## Abstract

For aerospace applications, honeycomb sandwich panels may have small perforations on the cell walls of the honeycomb core to equilibrate the internal core pressure with external gas pressure, which prevent face-sheet/core debonding due to pressure build-up at high temperature. We propose a new form of perforation on the cell walls of honeycomb sandwich panels to reduce the influence of the perforations on the cell walls on the mechanical properties. In this paper, the high temperature mechanical properties of a new vented Ti-6Al-4V honeycomb sandwich panel were investigated. A vented Ti-6AL-4V honeycomb sandwich panel with 35Ti-35Zr-15Cu-15Ni as the filler alloy was manufactured by high-temperature brazing. The element distribution of the brazed joints was examined by means of SEM (scanning electron microscopy) and EDS (energy-dispersive spectroscopy) analyses. Compared to the interaction between the face-sheets and the brazing filler, the diffusion and reaction between the honeycomb core and the brazing filler were stronger. The flatwise compression and flexural mechanical properties of the vented honeycomb sandwich panels were investigated at 20, 160, 300, and 440 °C, respectively. The flatwise compression strength, elastic modulus, and the flexural strength of the vented honeycomb sandwich panels decreased with the increase of temperature. Moreover, the flexural strength of the L-direction sandwich panels was larger than that of the W-direction sandwich panels at the same temperature. More importantly, the vented honeycomb sandwich panels exhibited good compression performance similar to the unvented honeycomb sandwich panels, and the open holes on the cell walls have no negative effect on the compression performance of the honeycomb sandwich panels in these conditions. The damage morphology observed by SEM revealed that the face-sheets and the brazing zone show ductile and brittle fracture behaviors, respectively.

## 1. Introduction

The honeycomb sandwich panels consisting of two thin face-sheets bonded to a lightweight honeycomb core are widely used in the engineering applications requiring high rigidity with lightweight [1,2,3,4,5,6]. Ti-6Al-4V alloy has excellent high temperature properties and good corrosion resistance, which is an ideal candidate for the face-sheets and core materials of honeycomb sandwich panels [7,8,9]. With low thermal conductivity, high stiffness to weight ratio, high strength to weight ratio and high temperature mechanical properties, Ti-6Al-4V honeycomb sandwich structures can be used in aerospace application, such as the thermal protection systems (TPS) components [10]. It can be operated at high temperature up to ~440 °C, compared to traditional Aluminum alloys (~300 °C) [11,12]. Hence, the investigation on the mechanical properties of the Ti-6Al-4V honeycomb sandwich structures has drawn much attention in recent decades.

In recent decades, numerous studies have focused on the mechanical behaviors of sandwich panels, such as compression [13,14,15,16,17], bending [18,19], and buckling [20,21,22] performance. For aerospace industries, it is necessary to understand the mechanical properties of honeycomb sandwich structures at high temperatures. A few investigations have been carried out on the mechanical behavior of honeycombs at different temperatures [12,23,24,25]. The interfacial bonding performance between the core and the face-sheets has a great influence on the mechanical properties of the sandwich structures [26,27]. Face-sheet/core debonding is a critical damage mode in sandwich panels. If the honeycomb sandwich panel is subjected to varying ambient pressure, temperature, ambient conditions, and if the cells are not vented, the pressure difference will result in mechanical stress in the core.

For aerospace applications, the honeycomb may have small wall perforations, which allows the entrapped air to escape, and equilibrates the internal core pressure with external gas pressure [28]. The hexagonal honeycomb with perforations on the cell walls is an alternative core material for the sandwich panel, which can carry load in out-of-plane direction and provide heat/fluid exchange channels by the perforations on the cell walls. Wang et al. [29] investigated the effects of perforation size, spacing and shape on the mechanical properties of honeycombs through the compression testing and finite element simulation. The vented honeycomb sandwich panel allows air to flow from cell to cell. However, the core walls include a series of holes will affect the integrity and mechanical properties of honeycomb sandwich panels. The traditional method is punch holes on all the cell walls of honeycomb core. In order to reduce the influence of the perforations on the cell walls on the mechanical properties, we propose a new form of perforations on the cell walls of honeycomb sandwich panels. Since the cell walls oriented in the L-direction is twice as thick as other cell walls, only punched holes on the cell walls along the L-direction of honeycomb core. This not only ensures the air escape of the honeycomb core, but also minimizes the number of holes on the honeycomb wall, so as to reduce the impact of holes on the mechanical properties of the honeycomb sandwich panels. Sometimes vented honeycomb sandwich panels are served as vital parts at high temperature, but we have no found the open literatures about the mechanical properties of vented honeycomb sandwich panels at high temperature. Therefore, the effect of high temperature on the mechanical properties of vented honeycomb sandwich panels is needs to be studied. The bending and flatwise compressive properties of a vented Ti-6AL-4V honeycomb sandwich panel have been studied at different high temperatures in this work.

In this paper, a new vented Ti-6AL-4V honeycomb sandwich panel was manufactured by high temperature brazing. The bending and flatwise compressive experiments of the vented Ti-6AL-4V honeycomb sandwich panel were carried out at room temperature, 160 °C, 300 °C and 440 °C, respectively. The flatwise compressive properties of the vented and unvented honeycomb sandwich panels were also compared. The microstructure of the brazed joint was observed by scanning electron microscope (SEM) and the element contents of the microstructure were measured by EDS. The failure modes of the damaged sandwich panels were identified, and the failure micromorphology was also observed by SEM.

## 2. Experimental Procedure

### 2.1. Metal Honeycomb Sandwich Panel Preparation

Vented Ti-6Al-4V honeycomb sandwich panels consisting of face-sheets of 0.3 mm in thickness and honeycomb core of 10 mm in height were manufactured in this work. The cell of the honeycomb core was a regular hexagon with each side of 3.23 mm in length. The chemical compositions of Ti-6Al-4V metal used in this study are given in Table 1, the contents of aluminum, vanadium, and titanium in the Ti-6Al-4V metal are about 6 wt.%, 4 wt.%, and 90 wt.%, respectively.

The manufacturing of the vented Ti-6Al-4V honeycomb sandwich panel was mainly divided into three steps. Firstly, the semi-hexagonal corrugated plate was rolled by 0.1 mm thickness metal strip which had holes of 0.5 mm in diameter. Secondly, the honeycomb core was manufactured with the corrugated metal plates by laser welding. Figure 1 shows the structure of the manufactured Ti-6Al-4V honeycomb core. The L-direction of the honeycomb core represents the direction of metal strip and the W-direction stands for the direction perpendicular to the metal strip. Therefore, the thickness of the cell walls oriented in the L-direction is twice as that of other cell walls. Note that the holes are punched on the middle height of the core and are along the L-direction. The third, the honeycomb core was brazed to the face-sheets with 35Ti-35Zr-15Cu-15Ni in the VAF-30 vacuum brazing furnace at 930 °C. The chemical compositions of 35Ti-35Zr-15Cu-15Ni brazing filler metal are summarized in the Table 2. Compared with the contents of Ti-6Al-4V metal, it can be observed that the filler metal does not contain Al and V elements and there are no Zr, Cu, and Ni in the honeycomb core and face-sheets. Figure 2 displays the VAF-30 vacuum brazing furnace and the manufactured metal honeycomb sandwich panel. The cell gaps and the interface between honeycomb core and face-sheets are well filled with the melted brazing filler metal, hence high interface bonding performance between the honeycomb core and the face-sheets is acquired.

### 2.2. Flatwise Compression Test

To evaluate the influence of temperature on the mechanical properties of the vented Ti-6Al-4V honeycomb sandwich panel, flatwise compression tests of the sandwich panels were conducted using the servo-hydraulic universal testing machine INSTRON 8801 of Illinois Tool Works Inc located in Glenview, IL, USA with ceramic heat insulation box at room temperature (20 °C), 160, 300, and 440 °C, respectively. Figure 3 shows the experimental set-up.

Test specimens were prepared according to the ASTM C365-2003 standard, the dimensions of the specimens were 50 mm × 25 mm × 10.6 mm, the cross-head speed was 1 mm/min during the test process. Five specimens were tested at each temperature.

The flatwise compression strength of the sandwich panel is determined by
(1)$$\sigma =\frac{P}{A}$$
where *P* and *A* are the maximum force before failure and the cross-sectional area, respectively.

The flatwise modulus of elasticity is given by
(2)$$E=\frac{t\Delta P}{A\Delta \mu }$$
where *t* is the thicknesses of the core. Δ*P* is the force difference between the two endpoints of the initial linear phase on the force-displacement curve. Δ*μ* represents the displacement difference corresponding to the Δ*P*.

### 2.3. Three-Point Bending Test

The three-point bending tests of the vented metal honeycomb sandwich panels were also carried out using the servo-hydraulic universal testing machine INSTRON 8801 (the servo-hydraulic universal testing machine INSTRON 8801 with ceramic heat insulation box at 20, 160, 300, and 440 °C, respectively. Test samples were fabricated according to the ASTM C393/C393M-11 standard. The three-point bending samples were divided into two types, one type was the L-direction samples with the L-direction of the core perpendicular to the cross-head, another type was the W-direction samples with the W-direction of the core along the span direction. The dimensions of the samples were 100 × 25 × 10.6 mm, the support span was 80 mm, and the cross-head speed was 0.5 mm/min during the test process. Five specimens were tested at each temperature.

## 3. Results and Discussion

### 3.1. Microstructure of the Brazed Joint

To evaluate the interface bonding performance between the honeycomb core and the face-sheets, the cross-sectional microstructure of the brazed joint in the double cell wall region was observed through SEM back-scattered electron images (BEIs), the result is shown in Figure 4a. It can be observed that there are no cracks or voids in the interface bonding regions which can cause disbonding failure during operation. The melted Ti-based braze metal provides good wettability, forming meniscus-like fillets at the interface between the cell walls of core and the face-sheet.

The face-sheet and the two cores formed dark phase in the SEM-BEI image due to the enrichment of lighter elements (Ti, Al, and V), and the braze material formed a bright phase with heavier elements (Zr, Ni, and Cu). A grey phase grows at the interface of the dark phase and bright phase as the diffusion effect during the heating process. In the gap between the two adjacent cell walls, the melted braze material climbs up and formed the filler due to the capillary action.

The microanalysis of EDS line scanning was performed to reveal the distribution of various elements present in the brazed joint. The major elements across the brazed joint measured by the EDS line scanning are shown in Figure 4b. The results corroborate the diffusion of Al, Ti, and V towards the filler as well as diffusion of Cu, Ni, and Zr towards the Ti-6Al-4V matrix.

The EDS chemical compositions obtained from the selected locations marked in Figure 4 are summarized in Table 2. As indicated in Figure 4, six different regions marked with 1, 2, 3, 4, 5 and 6 were examined. The regions marked with 1 and 3 represent the Ti-6Al-4V melting zone of the face-sheet and the honeycomb core, respectively. Region 2 stands for the interface between the Ti-6Al-4V matrix of the face-sheet and the 35Ti-35Zr-15Cu-15Ni brazing filler. The region 4 stands for the interface between the Ti-6Al-4V matrix of the honeycomb core and the 35Ti-35Zr-15Cu-15Ni brazing filler. The regions marked with 5 and 6 denote the melting zone of the 35Ti-35Zr-15Cu-15Ni brazing filler metal. Region 1 mainly includes a Ti-rich phase coupled with Al and V, which is consistent to the matrix Ti-6Al-4V metal. The element contents in the region 3 basically coincide with the matrix, while a small amount of Zr, Ni, and Cu have been inspected. At the interface region 2 and 4, the Zr, Ni and Cu elements from the melted 35Ti-35Zr-15Cu-15Ni brazing filler have diffused into the matrix during brazing process. Including a small quantity of Al and V, the element contents detected in region 5 are relatively in line with the brazing filler metal. A grey phase surrounded by the brazing filler metal is found in region 6, resulting from the reaction between the matrix and the brazing filler. The Ti content in region 6 is lower than that in the matrix metal, but higher than that in the brazing filler metal. Furthermore, the contents of Zr, Cu, and Ni of the grey phase are lower than that in the brazing filler.

Based on the contents of Zr, Cu and Ni in the regions marked with 1 and 3, it can be observed that more elements of the brazing filler metal diffuse into the cell walls of honeycomb than the face-sheet. Furthermore, the Al and V elements inspected in the 5 regions reveals that the elements of the matrix metal also diffuse into the brazing filler metal. The contents of Ni and Zr in region 2 are larger than region 1, and this indicates a higher tendency of diffusion from brazing filler to the cell wall of core than the face-sheet. The element contents in regions 2, 4m and 6 indicate that a reaction between the matrix Ti-6Al-4V metal and the 35Ti-35Zr-15Cu-15Ni brazing filler has occurred during the brazing process, which has a positive effect on the interface bonding performance between the honeycomb core and the face-sheet.

### 3.2. Flatwise Compressive Properties of the Vented Honeycomb Sandwich Panels

The flatwise compression properties of the vented metal honeycomb sandwich panels were investigated at 20, 160, 300, and 440 °C, respectively. The typical stress–strain curves of the flatwise compression tests are plotted in Figure 5, and the flatwise compression properties strength and elastic modulus calculated based on the experimental data are summarized in Table 3.

As shown in Figure 5, the flatwise compression strength and elastic modulus of the vented honeycomb sandwich panels decrease with the increase of temperature. The compression strength of the sandwich panels is 38.29MPa, 36.2MPa, 32.47MPa and 27.63MPa at 20 °C, 160 °C, 300 °C and 440 °C, respectively. Compared to the strength at room temperature, the compression strength of sandwich panels decreases by 5.5%, 15.2%, and 27.8% at 160, 300, and 440 °C, respectively. The compression elastic modulus of the sandwich panel is 943.87 MPa at 20 °C, which has a decrease of 30.3% to MPa at 160 °C, 35% to MPa at 300 °C, and 56.4% to MPa at 44 0 °C, respectively. Furthermore, the decrease of elastic modulus is larger than the strength for the flatwise compression of the vented honeycomb sandwich panels with temperature increasing.

To evaluate the influence of the holes on the flatwise compression properties of the sandwich panels in this work, the experimental tests of the unvented honeycomb sandwich panels were also conducted at 20, 160, 300, and 440 °C, respectively. Figure 6 displays the flatwise compression stress-strain curves of the unvented sandwich panels, and the compression strength and elastic modulus are given in Table 4. It can be found that the flatwise compression properties of the unvented sandwich panels are also decreased as the temperature increases. Compared with the experimental data of the unvented sandwich panels, there are small changes for the flatwise compression properties of the vented sandwich panels, and the difference is within 4%. Therefore, the influence of the holes on the flatwise compression performance of the honeycomb sandwich panels can be neglected in this study.

The damage morphology of the vented and unvented honeycomb sandwich panels subjected to flatwise compression is plotted in Figure 7; Figure 8, respectively. It can be observed that the compression bucking failure of the honeycomb core occurs at about one-quarter the thickness of the cell walls, while no damage is found near the holes. The results show that the compression performance and failure mode of vented honeycomb sandwich panels are same to the unvented honeycomb sandwich panels, indicating that the holes punched on the honeycomb core did not lead to the decrease of flatwise compression performance.

### 3.3. Three-Point Bending Properties of the Vented Honeycomb Sandwich Panels

To assess the influence of high temperature on the three-point bending performance of the vented honeycomb sandwich panels, the L-direction and W-direction sandwich panels were investigated at room temperature, 160, 300, and 440 °C, respectively. The typical load–deflection curves of the vented honeycomb sandwich panels are plotted in Figure 9.

The load–deflection curves of the L-direction and W-direction sandwich panels generally show a bilinear behavior at 20 °C. There is a sudden load drop when the load reaches the maximum, which indicates that a brittle failure has occurred for each type of sandwich panel. It can also be observed that the maximum load of the L-direction panel is larger than that of the W-direction panel, while the deflection of the L-direction panel is smaller than that of the W-direction panel. The horizontal segment of the load-deflection curves reveals the appearance of plastic deformation for both types of sandwich panels at 160 °C. The load also has a sudden drop when reaching the maximum, which indicates the occurrence of a brittle fracture behavior as well. The maximum load and the deflection of the L-direction sandwich panel both are larger than those of the W-direction sandwich panel. For each type of sandwich panel at 160 °C, the deflection only has about 10% increment relative to the experimental results recorded at 20 °C, while the maximum load of the L-direction and W-direction sandwich panels shows a decrease of 18.4% and 21.2%, respectively.

The sandwich panels show a brittle failure at 300 °C. Compared to the load-deflection of the L-direction sandwich panel, an obvious horizontal segment is recorded for the W-direction sandwich panel. Compared with the maximum loads at 20 °C, the maximum load of the L-direction and W-direction sandwich panels decreases by 40.1% and 33.4%, and the deflection increases by 20.2% and 19.4%, respectively.

It is apparent that each of the sandwich panels shows large plastic deformation, the ductile failure has occurred at 440 °C. The peaks visible has occurred in the curve of Figure 9d for the L-direction vented honeycomb sandwich panels. The difference of the maximum load between the L-direction sandwich panel and the W-direction sandwich panel is only about 9.2%. Compared to the 5 mm of the W-direction sandwich panel deflection, the value can reach about 6 mm for the L-direction sandwich panel. Compared with the results at 20 °C, the maximum load of the L-direction and W-direction sandwich panels decreases by 46% and 43.3%, and the deflection increases by 249.7% and 211.6%, respectively.

The failure modes of the vented honeycomb sandwich panels are illustrated in Figure 10. There are mainly six different failure modes observed for the honeycomb core, face-sheets, and the interface between them. The core cracking changes to buckling for the honeycomb core with temperature increasing. The upper face-sheet wrinkling, the lower face-sheet cracking, and the face-sheets buckling are observed for the face-sheets at different temperatures. The debonding between the honeycomb core and the face-sheets is also found for the bending sandwich panels.

The flexural properties and failure modes of the vented honeycomb sandwich panels are summarized in Table 5. It can be seen that the flexural strength decreases, and the deflection increases with temperature. Moreover, the flexural strength of the L-direction sandwich panels is larger than that of the W-direction sandwich panels at the same temperature. Based on the mechanical response and damage morphology of the sandwich panels, the damage evolution of the sandwich panels can be reconstructed at different temperatures. The face-sheets and the honeycomb core have a high strength and show low ductility at room temperature, the sandwich panels mainly show a brittle fracture behavior, tensile fracture of the lower face-sheet occurs when the tensile strength of the Ti-6Al-4V metal is reached. Though the vented honeycomb sandwich panels still show a brittle fracture behavior at 160 °C, the face-sheet bucking and delamination between the core and the face-sheets are found. As the testing temperature rises, the compression buckling of the upper face-sheet occurs due to the plastic deformation of the Ti-6Al-4V metal. Furthermore, the interface stress is high as the deformation between honeycomb core and the upper face-sheet is inconsistent. The delamination between the upper face-sheet and honeycomb core appears when the interface bonding strength is reached in the L-direction sandwich panels. When the temperature rises to 300 °C, the ductile fracture of the vented sandwich panels can be observed. The Ti-6Al-4V metal can deform largely owing to the improvement of plastic yielding. The honeycomb core of the W-direction panels locally yields and then collapses, resulting in the failure of the sandwich panels. Including slight honeycomb core compression buckling, the debonding between honeycomb core and the lower face-sheet is formed in a large area for the L-direction sandwich panels. Both types of sandwich panel failed, and this is attributed to the yielding and crushing of the honeycomb core and the face-sheet under the cross-head at 440 °C.

The micromorphology of the damaged W-direction sandwich panels at 20 °C is shown in Figure 11. Figure 11b is the partial enlarged drawing of the brazing region between the core and the brazing filler. The micromorphology figures show that the cleavage fracture occurs in the regions of the brazing filler, resulting from the brittle mechanical behavior of the brazing filler metal. Dimple fracture is observed in the regions marked with face-sheet and honeycomb core, which indicates the ductile failure of the Ti-6Al-4V metal. The size and depth of dimples in the honeycomb core are smaller than that of the face-sheet, which shows the more plastic deformation of the face-sheet than the honeycomb core. The diffusion zone has mixed-rupture characteristics of quasi-cleavage and dimples. This is due to the diffusion between the Ti-6Al-4V matrix and brazing filler. The debonding between the honeycomb core and the face-sheets occurs for the sandwich panels at 160 °C and 300 °C, and the micromorphology of the delamination is also plotted in Figure 12. It can be observed that the temperature has no significant influence on the damage morphology, and a large number of cleavage planes with a fan shape are formed at the debonding interface.

As shown in Figure 13, the honeycomb panel has no plastic deformation observed at the test temperature of 160 °C (a), a small plastic deformation at 300 °C (b), and a large plastic deformation at 440 °C (c) before the collapse of structure in the bending test.

When the honeycomb sandwich panel bears the bending load, the bending load is applied on the outer surface of the face-sheet, causing the stress in the face-sheets. The core layer mainly bears the sheer force and transmit the sheer force from one face-sheet to the other [30,31]. Under the same deformation, both the bending force on the face-sheets layer and the shear force on the core layer decrease with the decrease of the Young’s modulus at a higher temperature. At the same time, the plastic deformation of material can release more stress of the structure. The cooperative deformation ability of the face-sheets and core increases with the increase of plastic deformation ability, reducing the tendency of debonding between the core and the face-sheets during operation.

However, the increase of the plastic deformation ability of the structure below 300 °C is limited. When the temperature reaches 440 °C, the plastic deformation ability of the material is great enough to change the failure mode of the structure from brittle into ductile. The failure modes of vent Ti-6Al-4V honeycomb sandwich panels at 300 °C or below are mainly face-sheet cracking and face-sheet/core debonding. Moreover, at 440 °C, the failure mode changes to overall buckling and local buckling at the loading position. This agrees with the bending performance of the vented sandwich honeycomb panels which did not have ductile failure till 440 °C. When the vented honeycomb sandwich panel is subjected to three-point bending test at 440 °C, local buckling occurs at the loading position during the whole deformation process, so the peaks visible on curve Figure 9d of the L-direction vented honeycomb sandwich panels.

In our previous study [25], we investigated the high temperature mechanical properties and failure modes of titanium alloy honeycomb sandwich panels by three-point bending tests. The results showed that the failure modes change, the flexural strength decreases, and the maximum deflection increases with the increasing testing temperature in the three-point bending tests. The vented and unvented honeycomb sandwich panels have similar bending properties, and the failure mode, strength and maximum deflection have similar trends with the increase of temperature. And the holes on the cell wall of honeycomb core have not been damaged. So, the holes punched on the honeycomb core have no effect on the failure mode and bending properties at high temperature.

## 4. Conclusions

In this paper, the flatwise compression and flexural mechanical properties of the vented honeycomb sandwich panels were investigated at 20, 160, 300, and 440 °C. The honeycomb core and face-sheets were made of Ti-6Al-4V metal, and they were brazed with 35Ti-35Zr-15Cu-15Ni brazing filler metal. The influences of high temperatures and honeycomb core orientation on the mechanical performance of the vented sandwich panels were analyzed. The following conclusions can be drawn:(1)The elements of the Ti-6Al-4V metal and the 35Ti-35Zr-15Cu-15Ni brazing filler metal diffuse to each other and react at the interface between the honeycomb core and the face-sheets during the brazing process, which makes a positive influence on the interface bonding performance of the vented metal honeycomb sandwich panels. Compared to the interaction between the face-sheets and the brazing filler, the diffusion and reaction between the honeycomb core and the brazing filler are stronger.(2)Compared to the flatwise compression properties of the vented honeycomb sandwich panels at 20 °C, the compression strength shows 5.5%, 15.2%, and 27.8% reduction, and the elastic modulus decreases by 30.3%, 35%, and 56.4% for sandwich panels at 160, 300, and 440 °C, respectively. Compression buckling failure of the honeycomb core occurs at about one-quarter the thickness of the cell walls. The holes punched on the honeycomb core have not led to the decrease of the flatwise compression performance of the Ti-6Al-4V honeycomb sandwich panels in this study.(3)The flexural strength of the vented honeycomb sandwich panels decreases with temperature increasing, while the deflection is increased. The flexural strength and deflection of the L-direction sandwich panels is larger than that of the W-direction sandwich panels at the same temperature. The sandwich panels show a brittle fracture behavior at lower temperature, the failure modes mainly include the face-sheet tensile breaking and the debonding between the honeycomb core and the face-sheets. The ductile failure of the sandwich panels occurs resulting from the compression crushing of the honeycomb core and the local buckling failure of the face-sheets at higher temperature. The plastic deformation ability of the material is great enough to change the failure mode of the structure from brittle into ductile at 440 °C.

## Figures and Tables

**Figure 1 materials-13-03008-f001:**
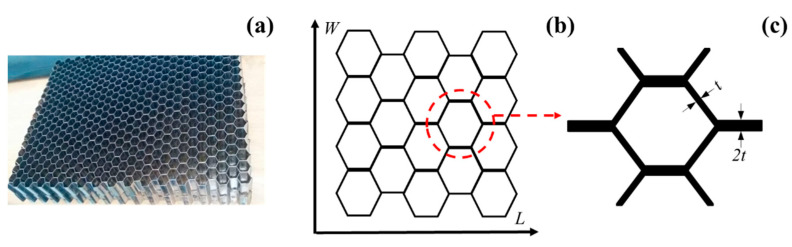
Ti-6Al-4V metal honeycomb core: (**a**) honeycomb core fabricated, (**b**) sketch diagram along the L and W directions and (**c**) cell unit.

**Figure 2 materials-13-03008-f002:**
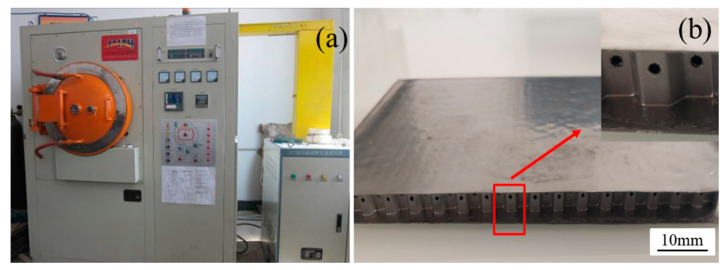
(**a**) VAF-30 vacuum brazing furnace and (**b**) vented metal honeycomb sandwich panel.

**Figure 3 materials-13-03008-f003:**
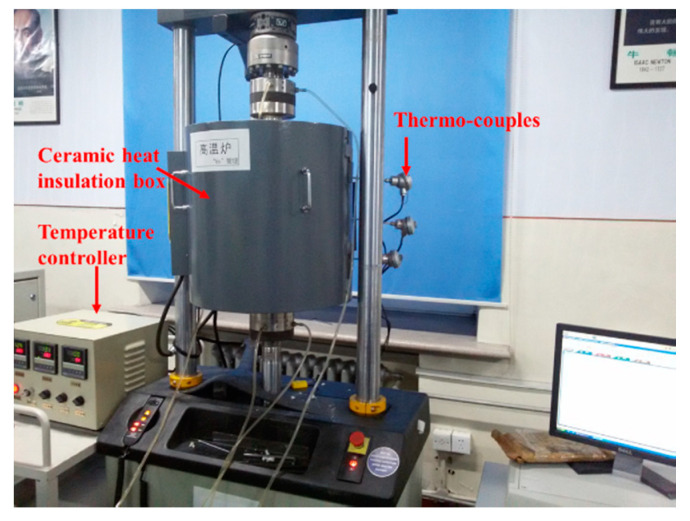
Experimental set-up.

**Figure 4 materials-13-03008-f004:**
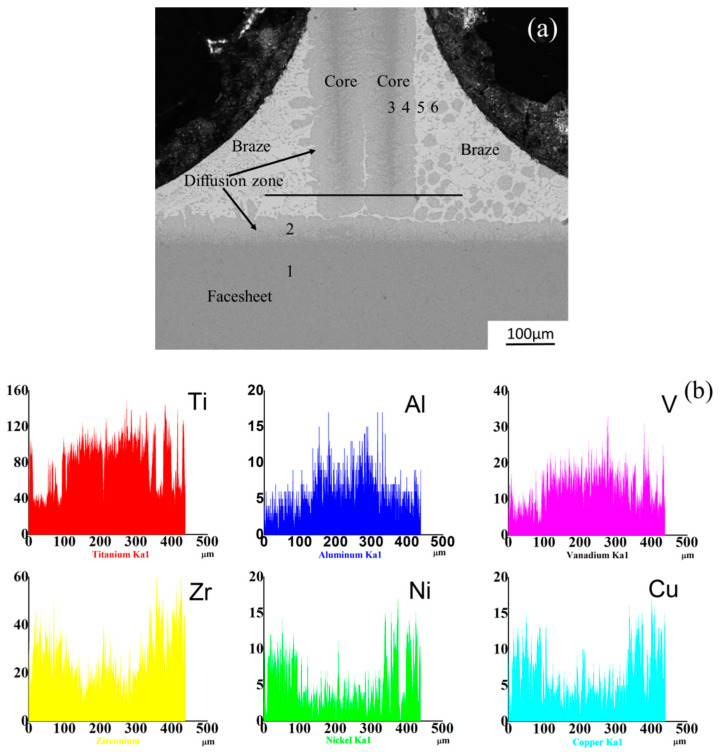
(**a**) The SEM-backscattered electron image (BSE) of the brazed joint. (**b**) Line (arrowed in Figure 6a) scanning of EDS showing the major elements distribution of brazed joint.

**Figure 5 materials-13-03008-f005:**
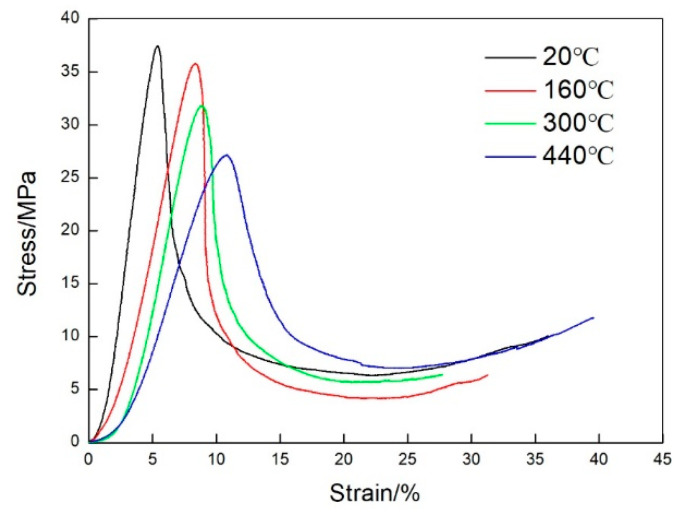
Flatwise compressive stress versus strain curves of vented honeycomb sandwich panels at high temperatures.

**Figure 6 materials-13-03008-f006:**
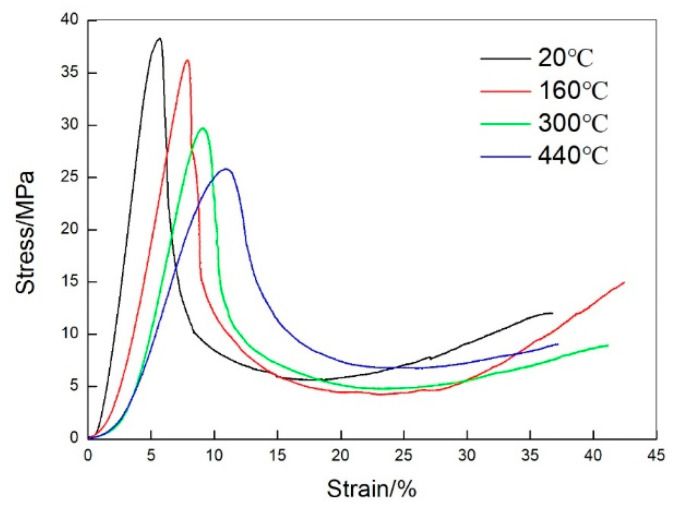
Flatwise compressive stress versus strain curves of the unvented honeycomb sandwich panels.

**Figure 7 materials-13-03008-f007:**
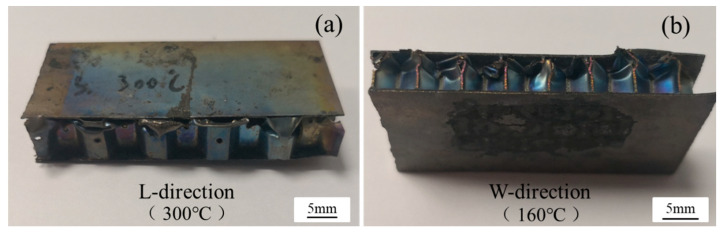
Damage morphology of vented honeycomb sandwich panels subjected to flatwise.

**Figure 8 materials-13-03008-f008:**
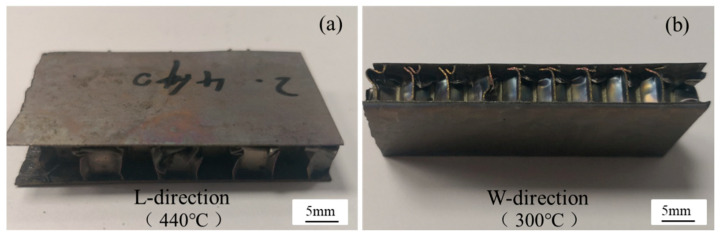
Damage morphology of unvented honeycomb sandwich panels subjected to flatwise.

**Figure 9 materials-13-03008-f009:**
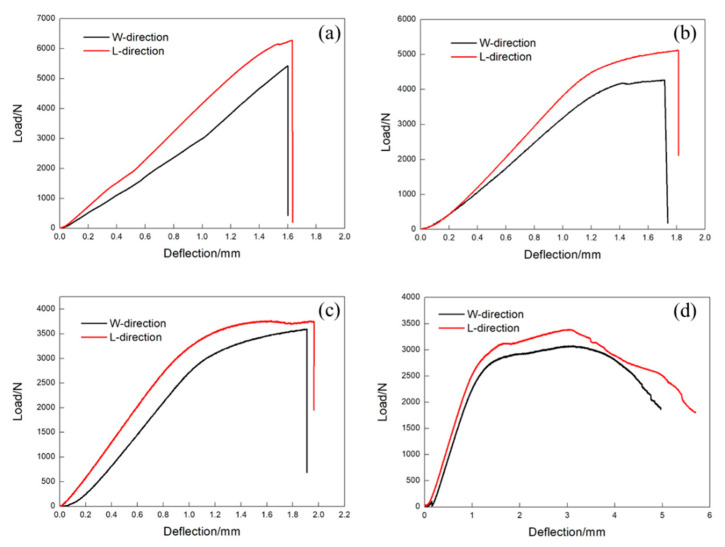
Typical load-deflection curves of the L-direction and W-direction sandwich panels at (**a**) 20 °C, (**b**) 160 °C, (**c**) 300 °C and (**d**) 440 °C.

**Figure 10 materials-13-03008-f010:**
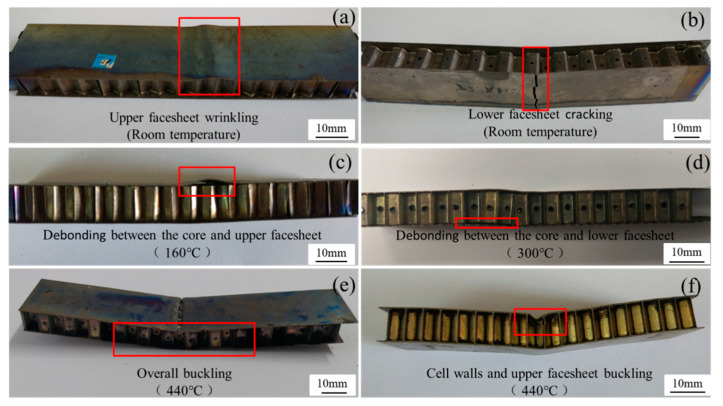
Failure modes of the vented honeycomb sandwich panels.

**Figure 11 materials-13-03008-f011:**
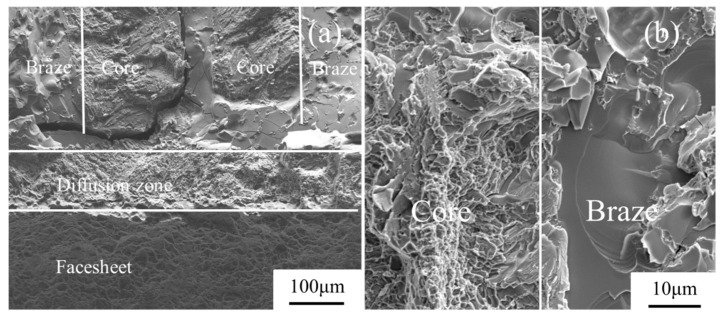
Micromorphology of the damaged W-direction sandwich panels at 20 °C: (**a**) the brazing joint region; (**b**) the magnified brazing region between the core and the brazing filler.

**Figure 12 materials-13-03008-f012:**
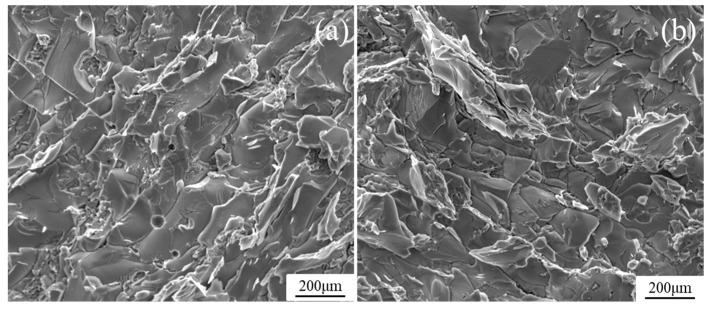
Micromorphology of the delamination for sandwich panels at (**a**) 160 °C and (**b**) 300 °C.

**Figure 13 materials-13-03008-f013:**
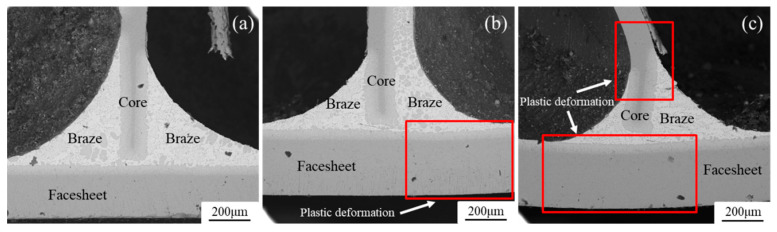
Microstructure of the brazed joint in the samples subjected to bending experiments at (**a**) 160 °C, (**b**) 300 °C, and (**c**) 440 °C.

**Table 1 materials-13-03008-t001:** Chemical compositions (wt.%) in the Ti-6Al-4V metal.

Al	V	Fe	Si	C	N	H	O	Ti
5.5–6.8	3.5–4.5	≤0.30	≤0.15	≤0.10	≤0.05	≤0.015	≤0.20	Bal. ^a^

(a): Residual chemical compositions (wt.%) in the Ti-6Al-4V metal.

**Table 2 materials-13-03008-t002:** SEM EDS chemical compositions (at.%) of the selected locations in Figure 4.

Zone	Al	Zr	Ti	V	Ni	Cu
1	6.43	0	87.86	5.71	0	0
2	10.61	0.76	78.03	7.68	1.51	1.51
3	12.24	1.06	81.72	2.39	1.32	1.26
4	4.51	15.12	62.03	4.6	4.58	9.15
5	3.31	24.28	37.95	1.67	14.91	17.88
6	2.69	17.85	57.04	1.41	8.11	12.9

**Table 3 materials-13-03008-t003:** Flatwise compression properties of the vented honeycomb sandwich panels.

	20 °C	160 °C	300 °C	440 °C
*σ*/MPa	38.29 ± 1.6	36.20 ± 1.5	32.47 ± 1.4	27.63 ± 1.2
*E*/MPa	943.87 ± 17	657.81 ± 15	613.97 ± 13	411.85 ± 13

**Table 4 materials-13-03008-t004:** Flatwise compression properties of the unvented honeycomb sandwich panels.

	20 °C	160 °C	300 °C	440 °C
*σ*/MPa	37.49 ± 1.5	35.80 ± 1.4	31.81 ± 1.5	27.16 ± 1.3
*E*/MPa	940.49 ± 16	634.61 ± 18	604.01 ± 14	397.67 ± 15

**Table 5 materials-13-03008-t005:** Flexural properties and failure modes of the vented honeycomb sandwich panels.

Temperature	Core Orientation	Defletion (Max)/mm	Load (Max)/N	Main Failure Modes
Room temperature	W	1.6	5427	upper face-sheet wrinkling; lower face-sheet cracking
	L	1.63	6279	lower face-sheet cracking; core cracking
160 °C	W	1.74	4276	upper face-sheet/core debonding
	L	1.81	5126	lower face-sheet cracking; upper face-sheet buckling
300 °C	W	1.91	3616	lower face-sheet/core debonding
	L	1.96	3764	core and the upper face-sheet buckling
440 °C	W	4.97	3076	core and the upper face-sheet buckling; overall buckling
	L	5.7	3389	core and the upper face-sheet buckling; overall buckling

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
