# Peer review of "High Temperature Mechanical Properties of a Vented Ti-6Al-4V Honeycomb Sandwich Panel"

_materials, 2020, doi:10.3390/ma13133008_

Round 1
Reviewer 1 Report
There are some other points to correct or to make the information more exact:
- One of the part of investigations presented in this manuscript is the changing the characteristics at flatwise compression. Authors found that the compression strength of vented sandwich panels decreases by 5.5% at heating from 20℃ to 160℃. Besides there are comparison of vented and unvented sandwich panels, where there are small changes for the flatwise compression properties of the vented sandwich panels, and the difference is within 4%. However there are not any values of accuracy of calculation or measurement errors of compression strength and compression elastic modulus. Are the errors compared with this changing or lower?
- There are some abbreviations in abstract SEM and EDS which are not explained here at the first mention in the abstract. The abbreviation CP for commercially pure titanium is used once (page 2 1st line) and isn’t explained. In this case it is better to use the phrase but not abbreviation
- Red letters in the figure 4 (e.g., Facesheet, Diffusion, Braze ) are not good readable.
- Some digits in data in the Table 4 are placed in the line lower. It is not suitable for reading.
- The numbers of the Load axis in the Figure 9b have only zeros. The digit of thousands and axis caption are cut. Then it is not possible to look the changing at 160o
- The part of 3.3 is devoted to investigations of three-point bending properties of vented honeycomb sandwich panels. What are these three-point bending properties for unvented honeycomb sandwich panels, if in previous part these types of sandwich panels are compared?

Author Response
We thank the Reviewer for the comments. All of the responses are given

Reviewer 2 Report
The article focuses on the mechanical properties of Ti-6Al-4V honeycomb sandwich panel at various temperatures. However, major revision will be required, especially to improve the readability and presentation of results. Here are some of the comments I have:
- The introduction is too detailed, especially the second paragraph which reviews previous work. That paragraph is over a page long and will need to be broken into shorter paragraphs for better readability. Some of the mentioned studies which are not necessarily important or relevant to this study can also be removed. It should be well-focused on the background of the work itself.
- Table 2 seems very redundant.
- I would suggest putting scale bars on any images of the sample (Figures 2b, 7, 8 and 10). The images in Figure 7 and 8 also seem low quality and not taken with good lighting, which makes it difficult to see.
- Figure 4 shows the interdiffusion of the filler into the matrix and could be easily shown by an EDX mapping of certain elements.
- Page 7, there seems to be a mistake referring to Figure 4 instead of Figure 5. (“As shown in Figure 4, the flatwise compression strength…”)
- The authors claimed that ductile failure was shown at 300degC, but the plot does not show such ductile behavior till 440degC.
- The 6 images in Figure 10 are not labeled with what their corresponding testing conditions.
- The comparison between the vented and unvented is stated as “almost no effect on the flatwise compression test” or “almost have no influence”. It seems very unscientific to say it has "almost" no effect. It is better to say they had similar compression performance where the vent played a very minimal role. You could also simply state the difference of compression performance, which is very small.
- Results have been presented but lack of any discussion on the mechanical behavior. Please have some discussion added, to discuss why the results are such. For example, why does the mechanical behavior transition from brittle to ductile at 300/440 and how does that compare to other systems? Why do you think the vents did not cause a difference in compression performance? Just a few examples, but please elaborate more on the results
Author Response
We thank the reviewer for the comments. All of the responses are given

Reviewer 3 Report
The presented paper concerns the influence of High Temperature Mechanical Properties of a Vented Ti-6Al-4V Honeycomb Sandwich Panel. The paper before publishing in the journal Metals the following issues should be corrected:
1. What is novelity in this paper, the autor schould be add information in the text, please you refine.
2. ,,The Ti-6Al-4V honeycomb sandwich panel can be operated at high-temperature” . How high the temperature, please specify, compare with traditional alloy.
3. ,,The manufacturing of the vented Ti-6Al-4V honeycomb sandwich panel was mainly divided into three steps. Firstly, the semi-hexagonal corrugated plate was rolled by 0.1mm thickness metal strip which had holes of 0.5mm in diameter. Secondly, the honeycomb core was manufactured with the corrugated metal plates by laser welding”. The third stage should be mentioned.
4. Figure 4, illegible description - I suggest you change the font color
5. What is caused by the peaks visible on curve d, please discuss in detail. Change of microstructure or chemical composition of the alloy at this temperature?
6. No detailed microstructure description particulary for 300 ℃ and 440C samples,
Author Response

(The authors gave the same response as above.)

Round 2
Reviewer 2 Report
The authors have made great efforts in the revision and it is suitable for publication.
Reviewer 3 Report
I accept the current version of the paper